# Sex-Specific Outcomes of Acute Stroke in Patients with Systemic Lupus Erythematosus: A National Inpatient Sample Study

**DOI:** 10.3390/jcm12020462

**Published:** 2023-01-06

**Authors:** Tiberiu A. Pana, Sona Jesenakova, Ben Carter, Rosemary Hollick, Mohamed O. Mohamed, Mamas A. Mamas, Phyo K. Myint

**Affiliations:** 1Keele Cardiovascular Research Group, Centre for Prognosis Research, Keele University, Stoke-on-Trent ST4 6QG, UK; 2Aberdeen Cardiovascular and Diabetes Centre, School of Medicine, Medical Sciences and Nutrition, University of Aberdeen, Aberdeen AB25 2ZD, UK; 3Institute of Applied Health Sciences, School of Medicine, Medical Sciences and Nutrition, University of Aberdeen, Aberdeen AB25 2ZD, UK; 4Department of Biostatistics and Health Informatics, Institute of Psychiatry, Psychology and Neuroscience, King’s College London, London SE5 8AF, UK; 5EULAR Centre of Excellence in Rheumatology, Aberdeen Centre for Arthritis and Musculoskeletal Health, University of Aberdeen, Aberdeen AB24 3FX, UK

**Keywords:** stroke, systemic lupus erythematosus, sex differences

## Abstract

**Background:** Systemic lupus erythematosus (SLE) is an autoimmune disorder associated with increased stroke risk. Its association with stroke outcomes remains poorly understood. In this study, we aimed to compare the sex-specific SLE-associated acute stroke outcomes. **Methods:** Stroke hospitalisations between 2015 and 2018 from the National Inpatient Sample were analysed. The associations between SLE and outcomes (inpatient mortality, length-of-stay > 4 days and routine discharge) were examined using multivariable logistic regressions, stratifying by sex and adjusting for age, race, stroke type, revascularisation, hospital characteristics and comorbidities. **Results:** A total of 316,531 records representing 1,581,430 hospitalisations were included. Median (interquartile range) age was 71 (60–82) years. There were 940 (0.06%) males and 6110 (0.39%) females with SLE. There were no associations between SLE and mortality amongst either females (odds ratio (95% confidence interval) = 1.11 (0.84–1.48)) or males (0.81 (0.34–1.94)). Nevertheless, SLE was associated with prolonged hospitalisation (1.17 (1.03–1.32)) and lower odds of routine discharge (0.82 (0.72–0.94)) amongst females. There were no associations between SLE and other adverse outcomes amongst males. **Conclusions:** The association between SLE and acute stroke outcomes was influenced by sex. While SLE was not associated with mortality in either sex, females with SLE had higher odds of prolonged hospitalisation and lower odds of routine home discharge compared to patients without SLE, while males did not exhibit this increased risk.

## 1. Introduction

Systemic lupus erythematosus (SLE) is a multisystem autoimmune condition with variable presentation and clinical manifestations, which can lead to significant morbidity and mortality [1]. Stroke is an important complication for the SLE patient population. In a recent systematic review and meta-analysis including 20 studies and 379,006 SLE patients, SLE was associated with a 2-fold increase in the risk of incident cardiovascular disease and stroke [2]. Similarly, a further meta-analysis including 40,652 SLE patients from five studies revealed 2-fold increased odds of incident ischaemic stroke and 80% increased odds of haemorrhagic stroke amongst SLE patients compared to the general population [3]. Furthermore, up to 11.8% of deaths of SLE patients may be attributable to cerebrovascular events [4].

In addition to the increased risk of stroke, previous studies report that SLE is associated with a higher inpatient mortality following both ischemic stroke (odds ratio = 2.18) and intracranial haemorrhage (odds ratio = 3.00) [5]. Furthermore, a meta-analysis including three observational studies found that SLE was associated with 68% higher stroke mortality [6]. However, these findings were derived from demographically different areas of the world, where management strategies of both SLE and stroke might differ significantly. For example, management of SLE appears to differ between Europe and Asia, where European practice is more likely to continue hydroxychloroquine and discontinue steroids in the management of patients in remission [7]. There is a need to examine the association between SLE and acute stroke outcomes in the patient population, reflective of contemporary U.S. practice. Finally, SLE is predominantly a female condition, with a female-to-male ratio of 9:1 [8]. Previous cohort studies [8,9] ascertained a higher overall mortality risk attributable to multiple different causes in males with SLE, where male sex was identified as a risk factor for death at the outset of the disease [9] and in the early disease course [10]. Hence, major sex differences exist in multiple aspects of the natural course of SLE. Nevertheless, sex differences in stroke outcomes amongst patients with SLE remain poorly described.

In this study of the National Inpatient Sample, we aimed to determine the sex-specific in-hospital outcomes of acute stroke in patients with comorbid SLE in contemporary U.S. practice. Furthermore, we also aimed to determine the sex-specific differences in the receipt of stroke revascularisation therapies associated with SLE amongst patients with ischaemic stroke.

## 2. Methods

The National Inpatient Sample is a publicly available database with no patient-identifiable information. Thus, ethical approval was not necessary for this project. The data supporting the findings of this study are available from the corresponding author upon reasonable request.

### 2.1. Data Source and Inclusion Criteria

The National Inpatient Sample (NIS) is the largest publicly available all-payer database, developed for the Healthcare Cost and Utilization Project (HCUP), which contains more than 7 million hospital stays annually in the United States [11]. The NIS records admission records representing a 20% stratified sample of all community hospital admissions in the United States in a given timeframe. Using the provided information on NIS strata, year of admission and sampling weights, estimates representative of ~97% of the U.S. population can be determined [12,13,14]. Prior to undertaking this project, all authors completed the Healthcare Cost and Utilisation Project (HCUP) Data Use Agreement Training Tool. All authors also read and signed the Data Use Agreement for Nationwide Databases.

Using data files containing annual admissions between 2015 and 2018, all records with a primary diagnosis of acute stroke (International Classification of Disease—Tenth Edition (ICD-10) codes I63.0–I63.9 for cerebral infarction, I61.0–I61.9 for intracerebral haemorrhage and I64 for undetermined stroke) were extracted. As no undetermined stroke cases (ICD-10 code I64) were present in the extracted sample, only acute ischaemic and haemorrhagic stroke admissions were included using these criteria. Furthermore, only hospitalisations between October 2015 and December 2018 were included due to a change to the coding of co-morbidities (ICD-9 to ICD-10) occurring after September 2015 [13].

Figure 1 details the study population flowchart. Of 327,741 records extracted from the NIS dataset with the primary diagnosis of acute ischaemic or haemorrhagic stroke admitted between October 2015 and December 2018, a total of 245 admissions with missing data on key variables and 11,210 elective admissions were excluded, resulting in a total of 316,531 hospitalisations being included in this study. Elective admissions were excluded to ensure that only those admissions caused by the acute stroke event were included and not any follow-up admissions. Having applied the sampling weights and excluded strata with single sampling units, included records were used to provide estimates for the population from which they were sampled, which included 1,581,430 admissions with the primary diagnosis of acute stroke.

To perform our secondary analysis—a comparison of the differences in the receipt of intravenous thrombolysis (IVT)/endovascular thrombectomy (ET) therapy between SLE and non-SLE groups—haemorrhagic stroke hospitalisations were excluded, yielding a study population of 1,402,150 hospitalisations with the primary diagnosis of acute ischaemic stroke (Figure 2).

### 2.2. Outcomes

The following primary outcomes were assessed: inpatient mortality, prolonged hospital stay >4 days and routine discharge, as well receipt of IVT/ET in patients with ischaemic troke. In-hospital mortality was ascertained using a standard NIS variable coding for vital status upon discharge (dead/alive) [15]. Prolonged hospital stay was defined as a stay of more than 4 days, based on expert clinical opinion and previous studies evaluating stroke outcomes among hospitalisations in the United States [16,17]. Routine discharge was ascertained using the provided discharge destination [18]. All discharges against medical advice and those discharged to an unknown destination were excluded from the analyses before applying sampling weights (n = 2982; 0.93%), allowing estimates for this particular outcome to be provided for 1,566,520 (99.06%) of acute stroke hospitalisations. Discharge destination was then dichotomised into routine home discharges and other discharges (“home health care”, “short-term hospital”, “other facilities including intermediate care and skilled nursing home”, and “died in hospital”). The “other discharges” category was subsequently used as a reference category in all analyses evaluating the discharge destination. A superunitary odds ratios resulted from the regression models assessing this outcome would therefore suggest higher odds of a favourable discharge outcome. For the secondary analysis, procedural ICD-10 codes were utilised to identify patients undergoing IVT (03CG3ZZ; 03CG4ZZ; 03CK3Z7; 03CK3ZZ; 03CK4ZZ; 03CL3Z7; 03CL3ZZ; 03CL4ZZ; 03CP3Z7; 03CP3ZZ; 03CP4ZZ; 03CQ3Z7; 03CQ3ZZ; 03CQ4ZZ) or ET (03CG3ZZ; 03CG4ZZ; 03CK3Z7).

### 2.3. Definition of Exposure and Confounders

Comorbid systemic lupus erythematosus (SLE) was the primary exposure of interest, identified using the respective ICD-10 codes to encompass all SLE subtypes: M320, M3210, M3211, M3212, M3213, M3214, M3215, M3219, M328, M329. SLE hospitalisations were then subdivided into 2 groups based on sex (female/male) and compared with non-SLE patients.

All models were adjusted for the following confounders: age, ethnicity, Elixhauser co-morbidities (congestive heart failure, valvular disease, pulmonary circulatory disease, peripheral vascular disease, paralysis, other neurological disorders, chronic pulmonary disease, diabetes, hypothyroidism, renal failure, arthropathies excluding SLE, liver disease, peptic ulcer disease, acquired immune deficiency syndrome, coagulopathy, obesity, weight loss, fluid and electrolyte disorders, anaemia, alcohol abuse, drug abuse, psychosis, depression and hypertension), previous history of cancer, haematological malignancies, other co-morbidities (dyslipidaemia, smoking, chronic obstructive pulmonary disease, sepsis, dementia, rheumatic heart disease, Parkinson disease, coronary heart disease, all-cause bleeding, infective endocarditis, pericarditis, pulmonary embolism, transient ischaemic attack, pulmonary hypertension, deep venous thrombosis, atrial fibrillation, pneumonia (including aspiration), shock, previous cerebrovascular disease), hospital bed size, location and teaching status. The co-morbidities were determined using the ICD-10 codes (Appendix A) or the HCUP Elixhauser co-morbidity software version 3.7 [19] and represent diagnoses assigned before or during the index stroke admission.

### 2.4. Statistical Analysis

All analyses were performed according to Healthcare Cost and Utilization Project guidelines [13], applying the provided discharge weights as probability weights and survey data analysis techniques stratifying by NIS stratum and year of admission [20] to account for patient clustering within hospitals while producing U.S.-wide estimates using Stata 16.0 [21]. A 5% threshold of statistical significance was used for all analyses (*p* < 0.05).

### 2.5. Descriptive Statistics

Patient characteristics were compared between patients without SLE, male patients with SLE and female patients with SLE. Independent-sample Kruskal–Wallis test and Pearson’s chi-squared test were employed to compare patient characteristics for non-normally distributed continuous and categorical variables, respectively. Whether a continuous variable was normally distributed was ascertained by visual inspection of the corresponding histogram.

### 2.6. Association between Systemic Lupus Erythematosus and In-Hospital Outcomes

Multivariable logistic regression models were employed to analyse the relationship between SLE and all in-hospital outcomes, stratifying by sex. Interaction terms between SLE and sex were also derived.

All models were adjusted for the stroke type (ischaemic/haemorrhagic), revascularisation therapy (thrombolysis, thrombectomy) receipt and the confounders listed above. A *post hoc* exploratory analysis was also undertaken to explore the interactions between SLE and ethnicity amongst females, adjusting for the confounders.

### 2.7. Secondary Analysis

Multivariable logistic regression models were employed to analyse the association between comorbid SLE and the odds of receiving IVT or ET therapy for acute ischaemic stroke, using the non-SLE hospitalisations as a reference group. All models were adjusted for sex and the confounders listed above.

## 3. Results

### 3.1. Descriptive Statistics

In 1,581,430 included admissions, the median (interquartile range) age was 71 (60–82) years, and the median (interquartile range) LOS was 3 (2–6) days. There were 940 (0.06%) male SLE and 6110 (0.39%) female SLE hospitalisations, with a median age between 60 and 61 years (Table 1). This was significantly lower compared to the 1,574,380 (99.55%) admissions in the non-SLE group with a median (interquartile range) age of 71 (60–82) years. The LOS was different among the three groups, with the median (interquartile range) LOS of 3 (2–6) days for non-SLE, 4 (2–6) days for male SLE and 4 (2–7) days for female SLE groups. Multiple differences in the proportion of non-SLE, male SLE and female SLE admissions were observed in terms of cardiovascular, respiratory and other comorbidities detailed in Table 1. Although there were no significant differences in hospital mortality (5.55%, 3.19% and 4.91% for non-SLE, males with SLE and females with SLE, respectively), marked differences were present among the three groups in terms of prolonged hospitalisation (35.91%, 39.89% and 41.73%) and routine home discharge (34.20%, 46.28% and 36.82%).

### 3.2. Statistical Analysis

#### 3.2.1. Association between Co-Morbid SLE and In-Hospital Outcomes amongst Males and Females with Acute Stroke

Table 2 and Figure 3 detail the adjusted logistic regression analysis characterising the association between the exposure groups and in-hospital outcomes. There were no significant associations between SLE and any of the studied outcomes amongst males. Amongst females, SLE was associated with prolonged hospitalisation (1.17 (1.03–1.32)), lower odds of routine discharge (0.82 (0.72–0.94)), but not in-hospital mortality (1.11 (0.84–1.48)). None of the interaction terms between sex and SLE reached statistical significance. The *post hoc* exploratory analysis did not reveal any statistically significant interactions between ethnicity and the associations between SLE and any of the in-hospital outcomes amongst females (Appendix A).

#### 3.2.2. Association between Co-Morbid SLE and IVT/ET Therapy Receipt in Acute Ischaemic Stroke (AIS)

Table 3 details the odds ratios underlying the association between co-morbid SLE (Non-SLE vs. SLE hospitalisations) and receipt of IVT/ET therapy for acute ischaemic stroke. No statistically significant differences were observed between the non-SLE vs. the SLE hospitalisations in terms of IVT/ET receipt.

## 4. Discussion

In the first study to explore the sex-specific relationship between SLE and acute stroke outcomes using a large, contemporary sample, we found significantly worse outcomes amongst females with SLE compared to patients without SLE. While there were no mortality differences, females with SLE were 17% more likely to be hospitalised >4 days and 18% less likely to be routinely discharged home. Nevertheless, there were no statistically significant differences in any of the in-hospital outcomes between males with SLE and the reference group. Given our small sample of males with SLE, these results must be interpreted with caution. Finally, for the study population with the primary diagnosis of acute ischaemic stroke, we found no statistically significant differences in receipt of revascularisation therapy between stroke patients with SLE and those without.

Stroke is a significant complication for patients with SLE [22,23,24]. A meta-analysis including 80,419 SLE patients from 10 population-based cohort studies found that patients with comorbid SLE have a twofold higher risk of ischaemic and a threefold higher risk of haemorrhagic stroke [22]. Similarly, a further meta-analysis found twofold increases in the risk of stroke associated with SLE [23].

This association between SLE and incident stroke may be explained by SLE-associated premature atherosclerosis [25]. Further lupus-related risk factors for stroke include anti-phospholipid antibody positivity and uncontrolled disease activity [26]. Despite this wide range of stroke risk factors in the SLE population, the acute-phase stroke management is currently the same as for the general population [26]. Our analyses suggest that this is indeed the case in real-world clinical practice, with no statistically significant differences demonstrable between the exposure groups in receipt of revascularisation therapy, after extensive adjustment for confounding factors.

Additionally, a previous systematic review and meta-analysis examining overall and cause-specific mortality in SLE patients found that these patients have a 68% higher risk of dying as a result of cerebrovascular accident when compared to the general population [6]. Similar results were also reported in a population-based study from Taiwan, which included 622 ischaemic stroke and 292 intracranial haemorrhage SLE patients admitted between 2000 and 2012, where SLE was associated with 2–3-fold higher inpatient mortality following stroke [5]. This is in contrast to our findings suggestive of no statistically significant differences in in-hospital mortality between stroke patients with SLE and those without in either sex. Nevertheless, given the differences in the study population demographics, factors such as ethnicity may partly explain these differences. Both studies [5,6] included patient cohorts prior to 2012, from multiple countries worldwide, whereas our findings reflect a more contemporary practice in the United States specifically, where the inpatient mortality of males and females with SLE does not seem to differ from the no-SLE group. Furthermore, our results add valuable insights on other important in-hospital outcomes, such as the routine home discharge and the length of hospitalisation, where female stroke patients with comorbid SLE have worse outcomes in both of these outcome measures of interest than their counterparts without SLE.

It was somewhat unexpected that SLE appeared to be associated with adverse outcomes in females, but not males, given that previous studies have shown that males with SLE tend to have more severe disease and worse outcomes compared to their female counterparts [27]. These results must, however, be interpreted in the light of the small sample of males with SLE included in our study. The different ethnic distributions between the males and females with SLE in our study may be a contributing factor, given that females with SLE had a higher proportion of non-white ethnicities, which appear more likely to have more severe SLE phenotypes [27]. We therefore performed a post hoc exploratory analysis to test this hypothesis. This, however, revealed no significant interactions between ethnicity and the SLE-associated excess odds for adverse stroke outcomes amongst females. This suggests that other important mediating factors such as stroke severity, treatment for SLE and hormonal/menopausal status may contribute to these sex differences. 

The results of our study have several important implications for clinical practice. Firstly, given the excess odds of adverse acute stroke outcomes in females with SLE, primary stroke prevention and patient education targeted at this group are of utmost importance, which has also been highlighted in prior studies [28]. Furthermore, given the wide range of factors predisposing SLE patients to stroke, such as lupus-related factors and cardiovascular risk factors, management strategies minimising these, such as SLE treatments to induce remission or achieve low disease activity [29] and primary cardiovascular preventative strategies, should be intensively pursued in order to decrease the incidence of SLE-associated stroke. Further research aiming to determine specific personalised management strategies for stroke patients with comorbid SLE is also warranted. As our findings highlighted the predisposition of females with SLE to disproportionately adverse acute stroke outcomes, targeted approaches aiming to reduce these sex disparities are also needed. Additionally, given the small number of males with SLE in our sample, the association between SLE and stroke outcomes in males also needs to be further examined including larger sample sizes. Finally, our study only explored short-term in-hospital stroke outcomes. Further long-term studies are therefore warranted to fully understand the impact of sex on acute stroke outcomes in patients with SLE.

Our study has several strengths. We included hospitalisations with primary diagnosis of acute ischaemic or haemorrhagic stroke admitted between late 2015 and 2018, yielding a study population representative of roughly 1.5 million admissions. Thus, with this large and contemporary study population, reflecting modern-day clinical practice, we were able to produce results representative of ~97% of the U.S. population [14]. Furthermore, our results are generalisable to populations with similar demographics to the United States, such as Australia or Western Europe, making them a valuable source of information for clinicians and researchers worldwide. Finally, our study is the first to provide sex-stratified acute stroke outcomes in a contemporary SLE patient population.

We also acknowledge limitations, mostly arising from the administrative nature of the included data. As a non-randomised study, our analyses were unable to eliminate residual confounding. The NIS does not provide information about the stroke aetiology beyond the distinction of ischaemic stroke vs. intracerebral haemorrhage, or the pre-stroke functional status of the included admissions. We also lacked information about stroke severity, such as the National Institutes of Health Stroke Scale. Similarly, we did not have any medication data and therefore could not adjust the analyses for either stroke preventative therapies, SLE disease-modifying therapies or drug triggers of SLE. Furthermore, we also lacked data on hormonal milieu, which may be a particularly important driver of sex differences in stroke outcomes. As our study is based on data from the United States, our results may not be applicable in middle- or low-income countries, where the disparities might be larger due to the possibility of less robust SLE management and stroke prevention strategies. Further studies are therefore needed to explore the sex-specific stroke outcomes in the SLE patient population in different parts of the world. Finally, given that comorbidity diagnoses were based on ICD-10 codes, we did not have any information regarding the time elapsed between the SLE diagnosis and the incident stroke. Similarly, the NIS does not have information on SLE management.

In conclusion, using a large, contemporary patient cohort from the U.S., we found that females with SLE, but not males with SLE, had an increased risk of poorer outcomes compared to patients without comorbid SLE, particularly in terms of hospitalisation length and routine home discharge. We also highlight for the first time that the association between SLE and acute stroke outcomes is influenced by sex. Further research should explore these associations further by examining outcomes such as long-term mortality and stroke recurrence.

## Figures and Tables

**Figure 1 jcm-12-00462-f001:**
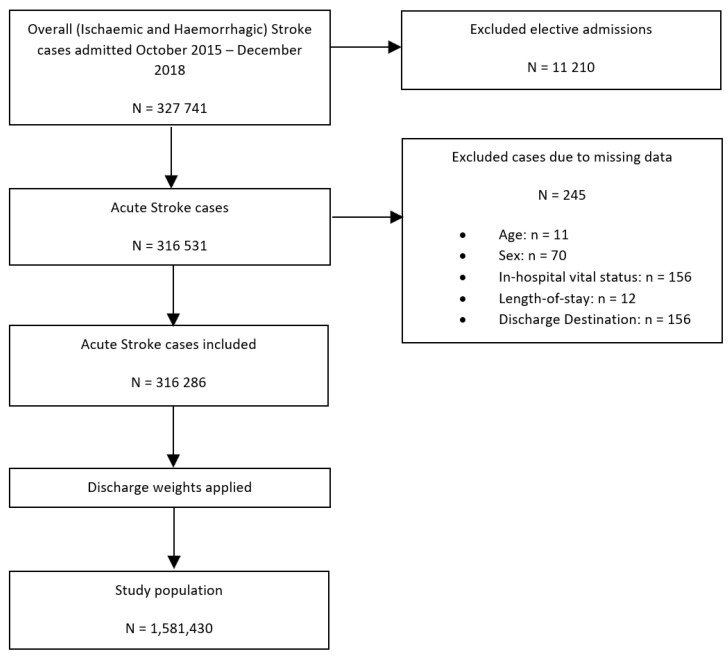
Patient population flowchart detailing the cohort included in the primary analysis assessing the association between co-morbid SLE and all in-hospital outcomes amongst patients with ischaemic and haemorrhagic stroke.

**Figure 2 jcm-12-00462-f002:**
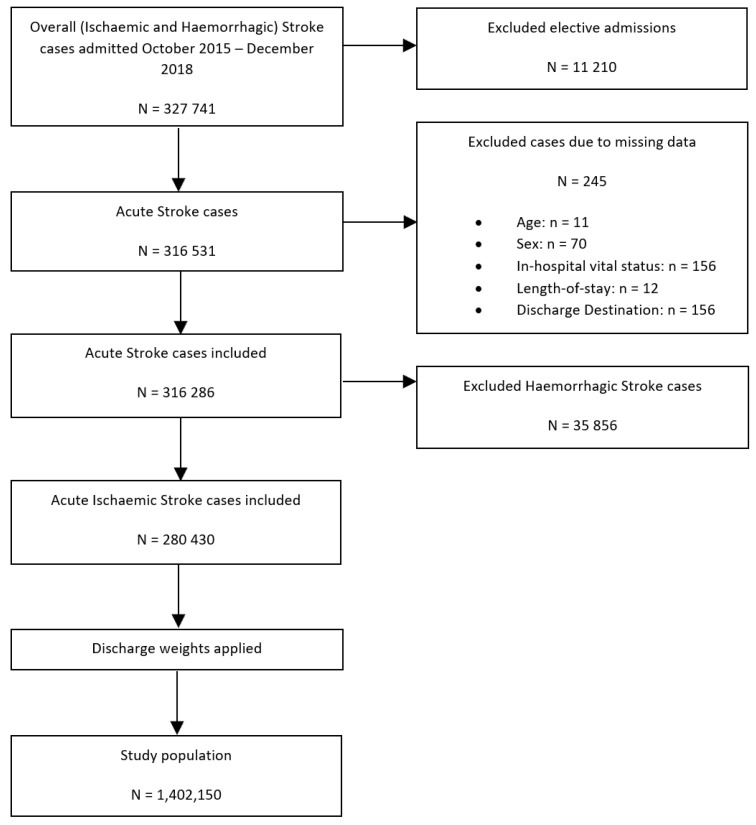
Patient population flowchart detailing the cohort included in the secondary analysis assessing the association between co-morbid SLE and IVT/ET receipt of intravenous thrombolysis or endovascular thrombectomy amongst patients with ischaemic stroke.

**Figure 3 jcm-12-00462-f003:**
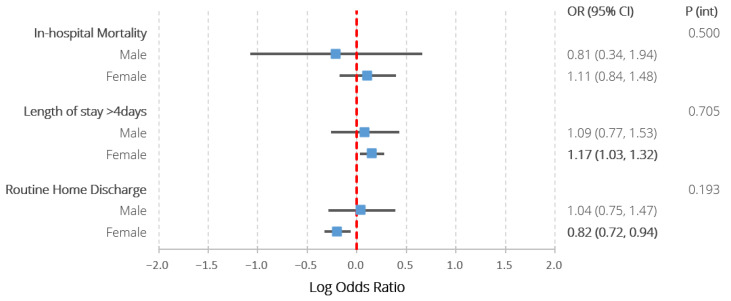
Results of multivariable logistic regression assessing the association between co-morbid SLE and all in-hospital outcomes amongst patients with ischaemic and haemorrhagic stroke, stratified by sex, also including the *p* values assessing the significance of interaction terms between SLE and sex. Models adjusted for age, ethnicity, hospital region, location and teaching status, stroke type, intravenous thrombolysis/endovascular thrombectomy receipt, previous cerebrovascular accident and a wide range of comorbidities (congestive heart failure, valvular disease, pulmonary circulatory disease, peripheral vascular disease, paralysis, other neurological disorders, metastatic cancer, chronic pulmonary disease, diabetes with chronic complications, diabetes without chronic complications, solid tumour without metastases, hypothyroidism, renal failure, arthropathies excluding SLE, liver disease, peptic ulcer disease, acquired immune deficiency syndrome, coagulopathy, obesity, weight loss, fluid and electrolyte disorders, deficiency anaemia, chronic blood loss anaemia, alcohol abuse, drug abuse, psychosis, depression, hypertension, previous history of cancer, lymphoma, dyslipidaemia, smoking, chronic obstructive pulmonary disease, sepsis, dementia, rheumatic heart disease, Parkinson disease, coronary heart disease, all-cause bleeding, infective endocarditis, pericarditis, pulmonary embolism, transient ischaemic attack, pulmonary hypertension, deep venous thrombosis, atrial fibrillation, pneumonia, aspiration pneumonia, shock). P (int)—*p* value for interaction term between sex and systemic lupus erythematosus. Statistically significant differences (*p* < 0.05) highlighted in bold.

**Table 1 jcm-12-00462-t001:** Baseline characteristics on admission of male and female stroke patients with and without comorbid SLE. Further descriptive statistics are detailed in Appendix A.

	All	No SLE	SLE-Male	SLE-Female	*p* Value
N (%)	1,581,430 (100)	1,574,380 (99.55)	940 (0.06)	6110 (0.39)	
Age, years, median (IQR) *	71.00 (60.00–82.00)	71.00 (60.00–82.00)	61.00 (48.00–70.00)	60.00 (48.00–71.00)	**<0.001**
Length of stay, days, median (IQR) *	3.00 (2.00–6.00)	3.00 (2.00–6.00)	4.00 (2.00–6.00)	4.00 (2.00–7.00)	**<0.001**
Stroke type, n (%) †					
Ischaemic	1,402,150 (88.66)	1,395,790 (88.66)	860 (91.49)	5500 (90.02)	0.152
Haemorrhagic	179,280 (11.34)	178,590 (11.34)	80 (8.51)	610 (9.98)	0.152
Ethnicity, n (%) †					
White	1,052,545 (66.56)	1,048,905 (66.62)	585 (62.23)	3055 (50.00)	**<0.001**
African American	273,450 (17.29)	271,235 (17.23)	250 (26.60)	1965 (32.16)	**<0.001**
Hispanic	113,340 (7.17)	112,695 (7.16)	40 (4.26)	605 (9.90)	**<0.001**
Asian	43,645 (2.76)	43,485 (2.76)	25 (2.66)	135 (2.21)	**<0.001**
Native American	6260 (0.40)	6230 (0.40)	<11 ‡	§	**<0.001**
Other	39,540 (2.50)	39,375 (2.50)	15 (1.60)	150 (2.45)	**<0.001**
Comorbidities, n (%) †					
Valvular disease	152,000 (9.61)	151,125 (9.60)	90 (9.57)	785 (12.85)	**0.001**
Peripheral vascular disease	147,340 (9.32)	146,665 (9.32)	150 (15.96)	525 (8.59)	**0.004**
Chronic pulmonary disease	248,220 (15.70)	246,845 (15.68)	175 (18.62)	1200 (19.64)	**<0.001**
Diabetes mellitus (without chronic complications)	283,560 (17.93)	282,605 (17.95)	140 (14.89)	815 (13.34)	**<0.001**
Diabetes mellitus (with chronic complications)	310,945 (19.66)	309,890 (19.68)	135 (14.36)	920 (15.06)	**<0.001**
Renal failure	261,820 (16.56)	260,325 (16.54)	215 (22.87)	1280 (20.95)	**<0.001**
Coagulopathy	70,360 (4.45)	69,580 (4.42)	135 (14.36)	645 (10.56)	**<0.001**
Obesity	211,175 (13.35)	210,050 (13.34)	135 (14.36)	990 (16.20)	**0.013**
Alcohol abuse	74,135 (4.69)	74,035 (4.70)	35 (3.72)	65 (1.06)	**<0.001**
Drug abuse	42,765 (2.70)	42,510 (2.70)	45 (4.79)	210 (3.44)	0.059
Hypertension	135,5140 (85.69)	1,349,435 (85.71)	815 (86.70)	4890 (80.03)	**<0.001**
Atrial fibrillation	392,825 (24.84)	391,640 (24.88)	180 (19.15)	1005 (16.45)	**<0.001**
Dyslipidaemia	892,075 (56.41)	888,955 (56.46)	460 (48.94)	2660 (43.54)	**<0.001**
Smoking	284,450 (17.99)	283,215 (17.99)	200 (21.28)	1035 (16.94)	0.310
Transient ischemic attack	11,240 (0.71)	11,185 (0.71)	<11 ‡	§	0.867
Rheumatic heart disease	44,050 (2.79)	43,780 (2.78)	45 (4.79)	225 (3.68)	**0.039**
Coronary heart disease	435,990 (27.57)	434,345 (27.59)	285 (30.32)	1360 (22.26)	**<0.001**
All-cause bleeding	280,530 (17.74)	279,360 (17.74)	145 (15.43)	1025 (16.78)	0.470
Pulmonary embolism	10,430 (0.66)	10,390 (0.66)	<11 ‡	§	0.546
Pulmonary hypertension	33,410 (2.11)	33,180 (2.11)	15 (1.60)	215 (3.52)	**0.002**
Infectious endocarditis	3400 (0.21)	3370 (0.21)	15 (1.60)	15 (0.25)	**<0.001**
Deep venous thrombosis	23,650 (1.50)	23,545 (1.50)	<11 ‡	§	0.506
Previous cerebrovascular disease	241,600 (15.28)	240,430 (15.27)	130 (13.83)	1040 (17.02)	0.207
Outcomes, n (%) †					
In-hospital mortality	87,715 (5.55)	87,385 (5.55)	30 (3.19)	300 (4.91)	0.233
Length of stay > 4 days	568,335 (35.94)	565,410 (35.91)	375 (39.89)	2550 (41.73)	**<0.001**
Routine discharge	541,095 (34.22)	538,410 (34.20)	435 (46.28)	2250 (36.82)	**<0.001**

SLE—systemic lupus erythematosus; IQR—interquartile range. Statistically significant differences (*p* < 0.05) highlighted in bold. * Independent-sample Kruskal–Wallis Test was used to compare differences amongst the 3 groups—No SLE, SLE-Male, SLE-Female—for this variable. † Pearson’s chi-squared test was used to compare differences amongst the 3 groups—No SLE, SLE-Male, SLE-Female—for this variable. ‡ Cell sizes ≤10 were not reported to avoid patient reidentification, according to the Healthcare Cost and Utilization Project guidelines. § Cell output suppressed as its value would allow identification of adjacent cell sizes ≤10.

**Table 2 jcm-12-00462-t002:** Results of multivariable logistic regression assessing the association between co-morbid SLE and all in-hospital outcomes amongst patients with ischaemic and haemorrhagic stroke, stratified by sex, also including the *p* values assessing the significance of interaction terms between SLE and sex.

All Stroke Types (Elective Admissions Excluded)			
		Odds Ratio (95% Confidence Interval)	*p* Value for Interaction (SLE × Sex)
In-hospital mortality	Males	0.81 (0.34–1.94)	0.500
	Females	1.11 (0.84–1.48)	
Length of stay > 4 days	Males	1.09 (0.77–1.53)	0.705
	Females	**1.17 (1.03–1.32)**	
Routine discharge	Males	1.04 (0.75–1.47)	0.193
	Females	**0.82 (0.72–0.94)**	

Models adjusted for age, ethnicity, hospital region, location and teaching status, stroke type, intravenous thrombolysis/endovascular thrombectomy receipt, previous cerebrovascular accident and a wide range of comorbidities (congestive heart failure, valvular disease, pulmonary circulatory disease, peripheral vascular disease, paralysis, other neurological disorders, metastatic cancer, chronic pulmonary disease, diabetes with chronic complications, diabetes without chronic complications, solid tumour without metastases, hypothyroidism, renal failure, arthropathies excluding SLE, liver disease, peptic ulcer disease, acquired immune deficiency syndrome, coagulopathy, obesity, weight loss, fluid and electrolyte disorders, deficiency anaemia, chronic blood loss anaemia, alcohol abuse, drug abuse, psychosis, depression, hypertension, previous history of cancer, lymphoma, dyslipidaemia, smoking, chronic obstructive pulmonary disease, sepsis, dementia, rheumatic heart disease, Parkinson disease, coronary heart disease, all-cause bleeding, infective endocarditis, pericarditis, pulmonary embolism, transient ischaemic attack, pulmonary hypertension, deep venous thrombosis, atrial fibrillation, pneumonia, aspiration pneumonia, shock). Statistically significant differences (*p* < 0.05) highlighted in bold.

**Table 3 jcm-12-00462-t003:** Results of multivariable logistic regression assessing the association between co-morbid SLE and receipt of intravenous thrombolysis or endovascular thrombectomy amongst patients with ischaemic stroke.

**Acute Ischaemic Stroke Cases**				
		**Non-SLE** **N = 1,395,840 (99.55%)**	**SLE** **N = 6310 (0.45%)**	** *p* ** **Value**
Intravenous Thrombolysis	N (%)	131,000 (9.39)	614 (9.75)	
	OR (95% CI)	1 (baseline)	0.92 (0.72–1.19)	0.534
Endovascular Thrombectomy	N (%)	42,350 (3.03)	204 (3.25)	
	OR (95% CI)	1 (baseline)	0.94 (0.66–1.36)	0.761

Models adjusted for age, ethnicity, sex, hospital region, location and teaching status, previous cerebrovascular accident and a wide range of comorbidities (congestive heart failure, valvular disease, pulmonary circulatory disease, peripheral vascular disease, paralysis, other neurological disorders, metastatic cancer, chronic pulmonary disease, diabetes with chronic complications, diabetes without chronic complications, solid tumour without metastases, hypothyroidism, renal failure, arthropathies excluding SLE, liver disease, peptic ulcer disease, acquired immune deficiency syndrome, coagulopathy, obesity, weight loss, fluid and electrolyte disorders, deficiency anaemia, chronic blood loss anaemia, alcohol abuse, drug abuse, psychosis, depression, hypertension, previous history of cancer, lymphoma, dyslipidaemia, smoking, chronic obstructive pulmonary disease, sepsis, dementia, rheumatic heart disease, Parkinson disease, coronary heart disease, all-cause bleeding, infective endocarditis, pericarditis, pulmonary embolism, transient ischaemic attack, pulmonary hypertension, deep venous thrombosis, atrial fibrillation, pneumonia, aspiration pneumonia, shock). SLE—systemic lupus erythematosus; OR—odds ratio; CI—confidence interval.

## Data Availability

The data supporting the findings of this study are available from the corresponding author upon reasonable request.

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
