# Peer review of "Sex-Specific Outcomes of Acute Stroke in Patients with Systemic Lupus Erythematosus: A National Inpatient Sample Study"

_jcm, 2023, doi:10.3390/jcm12020462_

Round 1

Reviewer 1 Report

In this interesting study on the potential relationship between acute stroke and SLE, the Authors investigated the impact of the differences between male and female in the incidence of acute stroke in patients with SLE. Specifically, stroke hospitalisations between 2015-2018 from the National Inpatient Sample (NIS) were analysed. The NIS is a publicly available database which contains more than 7 million hospital stays annually in the United States. The exposure groups were: no SLE (reference), males with SLE, and females with SLE.

In conclusion, the Authors highlighted for the first time that a relationship between SLE and acute stroke outcomes exists and it is strongly influenced by sex. Indeed, they reported that females with SLE, but not males with SLE, had an increased risk of poorer outcomes compared to patients without comorbid SLE, particularly in terms of hospitalisation length and routine home discharge. Moreover, they also investigated the potential relationship between SLE and risk to develop an ischemic or haemorragic stroke in male and in female patients with SLE. Although the study is interestig and extremely relevant, there are some aspects that need to be clarified.

Major points:

1.     In this study, the patients are divided into males with SLE and females with SLE whwerease in the the reference group the subjects are not distinguished according to sex. The authors have to explain this point in detail.

2.     How do the authors explain the worse prognosis of acute stroke in females with SLE? Please, indicate potential phenotipic or genotipic risk factors more evident in female than in male that might be responsible for these differences.

In this interesting study on the potential relationship between acute stroke and SLE, the Authors investigated the impact of the differences between male and female in the incidence of acute stroke in patients with SLE. Specifically, stroke hospitalisations between 2015-2018 from the National Inpatient Sample (NIS) were analysed. The NIS is a publicly available database that contains more than 7 million hospital stays annually in the United States. The groups analysed were: patients without SLE (reference), males with SLE, and females with SLE.

In conclusion, the Authors highlighted for the first time that a relationship between SLE and acute stroke outcomes exists and it is strongly influenced by sex. Indeed, they reported that females with SLE, but not males with SLE, had an increased risk of poorer outcomes compared to patients without comorbid SLE, particularly in terms of hospitalisation length and routine home discharge. Moreover, they also investigated the potential relationship between SLE and risk to develop an ischemic or haemorrhagic stroke in male and in female patients with SLE. Although the study is interesting and extremely relevant, some aspects need to be clarified.

Major points:

1.     In this study, the patients were divided into males with SLE and females with SLE whereas in the reference group the subjects are not distinguished according to sex. The authors have to clarify this point providing a detailed explanation on the sex of the components of reference group and the reasons why they did not considered this difference.

2.     How do the authors explain the worse prognosis of acute stroke in females with SLE? Please, indicate potential phenotypic or genotypic risk factors more evident in female than in male that might be responsible for these differences.

3. In the discussion the authors should formulate a hypothesis in relationship with the differences in the incidence of haemorrhagic and ischemic stroke between female and male with and without SLE. This aspect might improve the relevance and the impact of the study.

3.     It is necessary to better describe “Patient population flowchart – primary analysis” and “Patient population flowchart – secondary analysis”.

Minor points

It is necessary to correct minimal typographical errors present in the text of the paper.

Reviewer 2 Report

The precise mechanism of systemic lupus erythematosus is not known. It is known that lupus develops in genetically susceptible individuals after an additional stimulus, including hormonal factors and medications. Female  much more likely to develop the disease, due to the effects of sex hormones such as estrogen and prolactin; a higher risk of developing lupus or an exacerbation of the disease is also suspected in women taking hormone replacement therapy and oral contraceptives. Some medications can trigger so-called drug-induced lupus. Their list is long, among them are procainamide, chlorpromazine, methyldopa, hydralazine, isoniazid, TNF-alpha antagonists, interferon. Then the disease usually has a mild course, does not involve internal organs and regresses after discontinuation of the medication and sometimes steroid treatment is necessary.

It is advisable to add information on medications received, hormonal therapy. Analyze the impact of primary stroke prevention - whether patients were using anti-aggregation or anticoagulation drugs. 

Round 2

Reviewer 2 Report

I would like to appreciate the authors' response. 

I accept the paper in present form.